# The Development of a qPCR Assay to Measure *Aspergillus flavus* Biomass in Maize and the Use of a Biocontrol Strategy to Limit Aflatoxin Production

**DOI:** 10.3390/toxins11030179

**Published:** 2019-03-25

**Authors:** Alfred Mitema, Sheila Okoth, Suhail M. Rafudeen

**Affiliations:** 1Plant Stress Laboratory 204/207, Department of Molecular and Cell Biology, MCB Building, Upper Campus, University of Cape Town, Private bag X3, Rondebosch, Cape Town 7701, South Africa; 2Department of Botany, School of Biological Sciences, University of Nairobi, P.O. Box 30197, Nairobi 00100, Kenya; dorisokoth@yahoo.com

**Keywords:** *Aspergillus flavus*, aflatoxins, Pathogenesis, qPCR assay, maize lines, fungal biomass

## Abstract

*Aspergillus flavus* colonisation of maize can produce mycotoxins that are detrimental to both human and animal health. Screening of maize lines, resistant to *A. flavus* infection, together with a biocontrol strategy, could help minimize subsequent aflatoxin contamination. We developed a qPCR assay to measure *A. flavus* biomass and showed that two African maize lines, GAF4 and KDV1, had different fungal loads for the aflatoxigenic isolate (KSM014), fourteen days after infection. The qPCR assay revealed no significant variation in *A. flavus* biomass between diseased and non-diseased maize tissues for GAF4, while KDV1 had a significantly higher *A. flavus* biomass (*p* < 0.05) in infected shoots and roots compared to the control. The biocontrol strategy using an atoxigenic isolate (KSM012) against the toxigenic isolate (KSM014), showed aflatoxin production inhibition at the co-infection ratio, 50:50 for both maize lines (KDV1 > 99.7% and GAF ≥ 69.4%), as confirmed by bioanalytical techniques. As far as we are aware, this is the first report in Kenya where the biomass of *A. flavus* from maize tissue was detected and quantified using a qPCR assay. Our results suggest that maize lines, which have adequate resistance to *A. flavus,* together with the appropriate biocontrol strategy, could limit outbreaks of aflatoxicoses.

## 1. Introduction

Aflatoxins are categorised as an important class of mycotoxins that negatively affect human and animal health [1]. They are synthesised by several *Aspergillus* species; saprophytic fungi, that occurs widely, and grows on living and non-living substrates [2]. As an opportunistic fungal pathogen, *Aspergillus flavus* is able to infect many food crops, including cereals [3]. In developing countries, human exposure to aflatoxins is difficult to avoid during the various stages of processing and storage along the food chain [4]. Aflatoxin poisoning has also been reported in many parts of the world, in domestic and non-domestic animals, and other non-human primates [5]. Aflatoxins have been particularly problematic in eastern and central parts of Kenya, where there have been multiple outbreaks of aflatoxin poisoning among subsistence maize farmers [6]. Several cases of aflatoxicoses have been reported annually since 1981–2010, following consumption of maize contaminated with *A. flavus* and aflatoxins [6,7,8,9]. This problem has been particularly acute in Makueni county and the neighbouring region, which reported the highest number of cases and deaths in 2004, 2006, and 2010, due to aflatoxicoses [7,10,11].

The quantification of fungal biomass is critical in understanding the interactions between host plant susceptibility or resistance to a fungal pathogen, as well as identifying the competition between individual fungal species during disease progression [12]. Real time quantitative polymerase chain reaction (RT-qPCR) has been used to detect and quantify the fungal biomass in various plant host tissues [12,13,14]. Sanzani et al. [14] demonstrated that, the high level of sensitivity of qPCR enables the measurement of very low infection titres, which could correspond to the amount of pathogen present at the time of infection or during latent, non-symptomatic infections. Additionally, Miderosetal [15]. Developed and validated two RT-qPCR assays to estimate *A flavus* biomass in maize tissues using *Af2*, *Zmt3*, *INCW2-97* and *α-tubulin* marker genes, and studied the relationship between fungal biomass and aflatoxin accumulation. Mylroie et al. [16] moreover, developed a set of primers, and used RT-qPCR, to identify and quantify toxigenic, and non-toxigenic *A. flavus* strain’s fungal biomass on contaminated maize.

Biocontrol of toxigenic *A. flavus,* using atoxigenic strains, is already established with registered atoxigenic strains for the reduction of aflatoxin contamination on cultivated crops available to farmers [17,18,19,20,21]. The atoxigenic *A. flavus* (AF36) and *Afla*-Guard successfully suppressed aflatoxin producers on cottonseed in USA [21]. *Aflasafe^TM^* NG, has been provisionally registered for commercial use on maize in Nigeria [20]. Each atoxigenic strain is in a distinct vegetative compatibility group (VCG) to prevent hyphal anastomosis between the strains and affects aflatoxin levels by competitively excluding aflatoxin producers [17].

The use of atoxigenic *A. flavus* strains for biocontrol, is directly connected with methods for detecting aflatoxins, as it confirms that the biocontrol agent reduces aflatoxin contamination. The levels of aflatoxin in foods and feeds are strictly regulated [22,23]. This requires rapid, sensitive, quantitative, and relatively easy techniques for aflatoxin detection at various stages in the food chain [24]. Molecular techniques, based on PCR or culture-based methods, are used primarily to differentiate between aflatoxigenic and atoxigenic *Aspergillus* strains [25,26,27,28]. It must be noted that qPCR has been incorporated into protocols of the European Plant Protection Organization for the production, certification, and assessment of healthy plant materials [29,30]. With respect to bio-analytical techniques for aflatoxin detection, chromatographic methods such as high-pressure liquid chromatography (HPLC) is widely used in aflatoxin detection and is considered the gold standard for aflatoxin detection [24]. Aflatoxins are also detected by a thin layer chromatography (TLC), since most of the compounds fluoresce strongly under long-wavelength UV light [31,32,33]. HPLC, coupled with tandem mass spectrometry (HPLC-MS/MS), has become the method of choice, due to its high sensitivity and selectivity, which allows the determination of multiple mycotoxins in one sample [32,34].

This study investigated whether the respective maize lines KDV1, and GAF4, grown in different regions of Kenya, contribute to increasing or limiting the biomass of KSM014, an aflatoxigenic strain. In this regard we developed and tested a qPCR assay to quantify the amount of *A. flavus* biomass in infected maize tissues. Furthermore we tested if the atoxigenic *A. flavus* strain KSM012, (NCBI_accession MG385137) could serve as a biocontrol agent to minimise aflatoxin contamination of maize kernels colonised by the toxigenic *A. flavus* strain KSM014, (NCBI_accession MG385138). As far as we are aware, this is the first fungal qPCR biomass study on Kenyan *A. flavus* isolates from infected maize tissue. This approach could be used to discriminate between inbred maize lines, that are sensitive or resistant to specific *A. flavus* strains, and help understand the mechanism of maize defence response to *A. flavus* infection.

## 2. Results

### 2.1. Colonisation of Plant Yissues by A. flavus

The respective infection of maize lines KDV1 and GAF4 by *A. flavus* KSM014 resulted in changes in the maize plant phenotype. The infected kernels for both maize lines displayed stunted growth compared to the control kernels 14 days post infection (Figure 1 and Figure 4, Table 1). However, the KDV1 maize line had more severe symptoms when compared to the GAF4 line in terms of stunting of shoot and root growth (Table 1).

We wanted to confirm and support these phenotypic results for the two maize lines, by developing a RT-qPCR assay to detect and quantify *A. flavus* biomass load, in infected and control plants.

### 2.2. Gene Specificity and RT-qPCR Assays

The RT-qPCR primers designed for fungal and maize genes exhibited specificity as confirmed in control and infected tissues respectively (Figure 2). The MEP gene (203 bp), specific for maize, amplified in the control and infected plants for both maize lines (Figure 2). The MEP gene was also observed to be plant specific by the absence of cross-reaction with fungal gDNA. Similarly, both *Ef1a* (102 bp) and *β*-*Tub* (118 bp) were specific for *A. flavus* but the amplification of *Ef1a* appeared less sensitive in comparison to *β*-*Tub* (Figure 2). 

The *A. flavus β-Tubulin* (*β*-*Tub*) and maize membrane protein (*MEP*) primers were used to develop a RT-qPCR SYBR^®^ green assay to detect and quantify *A. flavus* gDNA in maize tissues. The *β*-*Tub* and *MEP* markers were found to have qPCR sensitivity, efficiency and linearity across a wide range of DNA concentrations of *R^2^* = 0.9997 for *β*-*Tub* and *R^2^* = 0.9988 for MEP respectively (Figure 3). 

The extracted fungal gDNA from co-infected shoots showed varied concentrations of fungal DNA compared to the roots according to 1-way ANOVA analysis and TMCT test (*p* < 0.05) (see Appendix A). 

The qPCR assay was performed 14 days post infection for both shoot and root tissue. No significant difference was seen in fungal biomass between the control and infected plant tissues for the GAF4 maize line (Figure 4a). In contrast, significant differences (*p* < 0.05) in fungal biomass for the KDV1 maize line was observed upon infection for both the root and shoot tissue (Figure 4b). The amount of fungal DNA was lower in the infected tissue of GAF4 maize line when compared to the KDV1 line.

### 2.3. In-Vitro Viocontrol Strategies in Aflatoxin Management and Aspergillus Flavus

We wanted to determine whether the atoxigenic *A. flavus* strain KSM012 (NCBI accession MG385137) could act as a biocontrol agent against the aflatoxigenic KSM014 strain, upon infection of maize kernels from two different lines. The atoxigenic and aflatoxigenic isolates were co-infected into KDV1, and GAF4 maize kernels, respectively followed by detection of aflatoxin production, using bio-analytical analyses.

The co-infected maize lines KDV1 and GAF4 had different rates of fungal colonisation following co-inoculation with *A. flavus* KSM012 and KSM014 (Figure 5). The maize kernels of the resistant (GAF4) and sensitive (KDV1) lines were co-infected with atoxigenic, and aflatoxigenic strains at different ratios (0:100; 25:75; 50:50; 75:25; 100:0), respectively. As expected, control plants inoculated with sterile water had no fungal growth, which indicated that there was no contamination by fungal spores during the experimental procedure (Figure 5f, controls). The KDV1 maize kernels had higher levels of colonisation by *A. flavus* compared to GAF4 maize kernels at a co-infection ratio of 50:50 (Figure 5c).

### 2.4. Aflatoxin Analyses after Biocontrol Strategy 

We wanted to determine whether the biocontrol strategy had any impact in reducing detectable aflatoxins, using TLC plate and HPLC techniques. The TLC plates examined and showed significant reductions in the aflatoxin levels from plants inoculated with a 50:50 ratio of atoxigenic to aflatoxigenic strains of *A. flavus* (Figure 6, lane 5). At 365 nm, visible spots with blue and green fluorescence for aflatoxins matching with the corresponding aflatoxin standards were observed (Figure 6, red and blue arrows).

At 254 nm, there was no observable blue or green fluorescence (data not shown). Thus, the presence or absence of aflatoxins and their derivatives could be visualised and identified at 365 nm. Staining with *p*-anisaldehyde, vanillin in phosphoric acid or iodine vapour did not yield any results (data not shown). HPLC chromatograms generated for the individual isolates had retention times for aflatoxins: AFG1-11.39-11.68; AFB1-12.72-12.84; AFG2-17.71-17.80; and AFB2-18.73-18.91 as observed in our previous work Mitema et al. [33].

As expected, the non-toxin producing strain (isolate KSM012) had neither an HPLC peak nor blue fluorescence on TLC plates (Figure 6, lane 4). Isolate KSM014 had peaks and fluorescence corresponding to AFB1 and AFB2, confirming that this isolate was aflatoxigenic as previously described by Mitema et al. [33]. Isolate KSM015, produced AFB1, AFB2, AFG1 and AFG2 confirming its previous identification as an SBG morphotype [33]. The detection limits and sensitivity, including linearity, showed that the method developed was acceptable for mycotoxin determination from cultures of *A. flavus.* The limit of detection (LOD) and limit of quantification (LOQ) ranged from 0.01–6.8 µg/mL and 0.02–35.81 µg/mL, respectively [33].

An analysis of the metabolites extracted from the co-infected cultures in the biocontrol strategy identified the HPLC peaks associated with aflatoxins (Figure 7A,B). The amount of AFB1 and AFG2 was significantly higher (*p* < 0.05) in GAF4 maize line (Figure 7A) than in the KDV1 at the co-infection ratio of 50:50. In the KDV1 maize line, aflatoxin contamination was reduced significantly when the plants were co-inoculated with the atoxigenic to aflatoxigenic isolate at ratios of 50:50 and 75:25 respectively (Figure 7B). GAF4 maize appeared less susceptible to *A. flavus* colonisation compared to KDV1 when looking at the concentration of the aflatoxins measured after co-infection (Figure 7).

## 3. Discussion 

GAF4 is a *Striga* spp. resistant maize line cultivated in Kisumu, Kibos, Homa Bay, and some parts of Nandi, while KDV1 is an open pollinated maize variety cultivated in Makueni and the neighbouring counties. KDV1′s increased susceptibility to *A. flavus* KSM014 infection could be a contributing factor to the previously reported frequent aflatoxicosis outbreaks and high levels of aflatoxin contamination for Makueni and the neighbouring regions [6,9,21,35]. These results are also consistent with our previous study, which showed that the majority of *A. flavus* isolates from Makueni produced high amounts of aflatoxin AFB1, AFB2 [33].

The phenotypic observations (Figure 1; Table 1) suggest that KDV1 maize line, grown in the Makueni county, is more susceptible to the *A. flavus* infection, whereas the GAF4 maize line grown in Kisumu and Homa bay counties appeared more resistant to the infection. The lower level of fungal DNA observed in the infected tissue of GAF4 maize line, compared to the KDV1 line, suggests that GAF4 was more resistant to *A. flavus* KSM014 infection (Figure 4; Appendix A) and this supports the previous phenotypic observation.

We developed a qPCR assay with the *A. flavus β*-*tub* gene and demonstrated that it can be used to quantify *A. flavus* biomass in both shoot and root tissue of different maize lines. The current study indicated that the *β*-*Tub* gene was a more suitable marker for the detection of *A. flavus* in maize tissues compared to *Ef1*αand was therefore used in fungal biomass determination. The respective marker genes chosen for *A. flavus* and maize were found to have qPCR sensitivity, efficiency and linearity across a wide range of DNA concentrations (Figure 2 and Figure 3). A critique of this approach is that fungal quantification using genomic DNA does not allow discrimination between viable and non-viable fungal biomass and quantification using qPCR assay [15,16]. 

We measured fungal biomass fourteen days after infection when symptoms of the infection were phenotypically visible. However, others have found that fungal biomass could be detected even before symptom development. Debode et al. [36] detected the presence of *Colletotrichum acutatum* by qPCR in strawberry leaves two hours post-inoculation even though the first disease symptoms appeared only after 96 h. Similarly, Divon & Razzaghian [37], could measure *Fusarium langsethiae* DNA in oats independent of the disease symptoms. Both findings demonstrated that fungal presence can be detected earlier, enabling the selection of resistant plants even when samples are indistinguishable based on visual assessment. 

The detection limits and sensitivity, including linearity of our results were similar and compared favourably to those of Gallo et al. [38] and Malachová et al. [32], who obtained LOD range of 0.6–1.9 µg/kg and a LOQ range of 0.02–0.05 mg/kg for aflatoxins extracted from highly contaminated animal feedstuff.

Our biocontrol strategy showed that aflatoxin production by the aflatoxigenic strain, KSM014, was inhibited by the atoxigenic strain, KSM012 at specific ratios. These observations suggest that upon colonisation of kernels by the aflatoxigenic isolate, the atoxigenic strain has the potential to limit colonisation of the aflatoxigenic isolate leading to inhibition or reduced aflatoxin levels. The reduction of aflatoxins could occur by the atoxigenic competitively excluding or displacing aflatoxigenic strains as suggested in other biocontrol studies [17,18,19,20,39,40]. Competitive exclusion by atoxigenic isolates results in reduced amount of aflatoxin during co-infection [41], a process aided by primary host contact. Huang et al. [42] showed that both the down-regulation of aflatoxin biosynthesis and variance in ability among fungal isolates to utilize nutrient resources could limit the amount of aflatoxin produced [43].

Biocontrol of aflatoxins by atoxigenic isolates is a cost-effective method for managing aflatoxins and could provide a long-term solution to aflatoxin contamination in developing countries, including sub-Sahara Africa [20]. The implementation of biocontrol strategies showed a reduction in aflatoxin contamination of peanuts and cereals by approximately 74.3% to 99.9% when coated with atoxigenic *A. flavus* strains [19,20]. These observations were similar to our results where aflatoxins were not detected or reduced [33,44]. Our results suggest that maize lines that have adequate resistance to *A. flavus* together with the appropriate biocontrol strategy may limit the possibility of aflatoxicoses outbreaks.

For biocontrol and biosafety applications of atoxigenic *A. flavus* isolates in the field, the atoxigenic *A. flavus* isolates should ideally be indigenous, genetically stable and belong to a VCG that contains no aflatoxigenic members [17,18,45]. Atehnkeng et al. [19] identified atoxigenic isolates (La3279, Og0222, Og0437 and Ka16127) as potential candidates for biocontrol in maize fields in Nigeria. Based on our study, KSM012 might be a candidate as part of a consortium of atoxigenic isolates for aflatoxin and *A. flavus* mitigation in Kenya and thus further study on this strain is warranted.

Medina et al. [46]; Lima et al. [47] and Stevenson et al. [47] all found that under laboratory conditions, the abiotic stress-related factors such as temperature, humidity, water activity and solutes can impact growth and aflatoxin production by *Aspergillus*. It is therefore important to first determine the impact of these environmental factors on aflatoxin production by *A. flavus* KSM102 under laboratory and field conditions to ensure its safe use and efficacy of as a biocontrol agent.

## 4. Conclusions

We found KDV1 maize line, which is cultivated in Makueni region and its environs to be more susceptible to *A. flavus* infection than GAF4 maize line grown in other regions of Kenya. This, susceptibility may be one of the possible reasons for the frequent cases of aflatoxicosis in Makueni than in Nandi, Kisumu and Homa Bay.

The *β-tub* gene is a potential marker for quantification of the *A. flavus* biomass load in maize plants compared to the *Ef1*αgene and both were specific for *A. flavus*. The MEP primers were specific for maize and had no cross contamination with *A. flavus* DNA. The specificity of the qPCR assay for *A. flavus* biomass quantification makes it potentially useful tool for screening of *A. flavus* maize lines for resistance to *A. flavus* and associated breeding strategies, identifying potential asymptomatic infections and help understand the mechanism of maize defence response to *A. flavus* infection.

We showed that a biocontrol strategy using the atoxigenic *A. flavus* isolate KSM012, was able to inhibit aflatoxin production by the aflatoxigenic *A. flavus* isolate KSM014 after co-infection of maize kernels at a 50:50 ratio. These findings are promising and might be suitable for future development of a biocontrol system, including more atoxigenic *A. flavus* isolates, appropriate for aflatoxin mitigation against aflatoxigenic *A. flavus* isolates in Kenya.

## 5. Materials and Methods

### 5.1. Cultures of Fungi

The aflatoxigenic *A. flavus* KSM014 isolate and other cultures were cultivated and maintained as described previously [33]. Two *A. flavus* isolates: KSM012 (atoxigenic strains) and KSM014 (aflatoxigenic strain) were grown on both aflatoxin inducing and non-inducing medium: Yeast Extract Sucrose (YES) and Yeast Extract Peptone (YEP) respectively and incubated in the dark for seven days at 30 °C. Conidia were harvested with cotton swabs and suspended in aqueous Tween 20 (0.2%). The working concentration was adjusted to spore suspension 1 × 10^6^ conidia/mL measured using a haemocytometer. The inoculum was stored at 4 °C and used within 1 week or stored as described earlier [33].

### 5.2. Maize Cultivars

Kenya Dry land Varieties KDV1 and GAF4 were purchased from Kenya Agricultural and Livestock Research Organisation (KALRO), Nairobi, Kenya. The varieties were selected based on the agroecological region in which they were cultivated and their drought tolerance. KDV1 is an open-pollinated variety that is recommended for low to medium altitude. It matures early, is drought tolerant and flowers between 45–52 days after germination. It is commonly grown in Makueni and Homa Bay (http://drylandseed.com). GAF4 is a *Striga* tolerant variety developed by the Kenya Agricultural Research Institute in Kibos, Kisumu county. It is grown in parts of western Kenya: Kisumu, Homa Bay and Busia [48].

### 5.3. Reagents and Media Preparation

Murashige and Skoog medium (MS), phytagel, glycine, nicotinic acid, thiamine hydrochloride, pyridoxine hydrochloride, myo-inositol, potassium hydroxide were purchased from Sigma-Aldrich (USA). MS vitamins; 250 mg nicotinic acid, 5 g myo-inositol, 500 mg pyridoxine-HCl, 100 mg glycine, and 500 mg thiamine-HCl was prepared in sterile water, filter sterilised and stored at −20 °C until used. The MS media was prepared by dissolving 2.15 g MS salts in sterile water, then 10 mL of MS vitamin stock was added and adjusted to the pH 5.7 with 1 M KOH, and the volume adjusted to 1 L with sterile water. Five grams of phytagel was added to MS media prepared and microwaved to dissolve the salts. Media (50 mL) was dispensed into tissue culture vessels, autoclaved, and allowed to cool in a biosafety cabinet (BSC) level 2 (Contained Air Solutions (CAS) BioMAT2, Pocklington York, UK) for approximately one hour before being inoculated. Potato dextrose agar, yeast extract, sodium chloride, ammonium acetate, tryptone, mycological peptone, malt extract agar, agar, chloroform, acetone, ethanol, methanol, dichloromethane, acetonitrile, formic acid (>98%) and trifluoracetic acid (99.8%) were from Sigma-Aldrich. Mycotoxin reference standards of aflatoxin B + G mixture dry concentrate containing 5.8 µg AFB1, AFG1 and 1.7 µg AFB2, AFG2/mL were from Sigma Aldrich (Darmstadt, Germany). Pure and ultrapure grade water was processed by Milli Q water purification system (Millipore LTD, Bedford, MA, USA). 

### 5.4. Seed Sterilisation and Aspergillus flavus Infection−

Twenty seeds were sterilised (in triplicates) by soaking in 20 mL of 95–100% ethanol for 1 min with brief shaking for 15 s. The ethanol was discarded and replaced with 20 mL of 2.5% sodium hypochlorite. The seeds were left to stand for 15 min, then shaken for 30 s and the liquid discarded. Seeds were washed 5 × with sterile water (20 mL) with intermittent shaking between each wash. A volume of 50 mL of sterile water was added and left to stand for 1 h at room temperature (RT). The water was replaced with 20 mL of 2% Tween 20, and shaken for 30 s. The seeds were inoculated (in triplicates) with 20 mL conidia suspensions (1 × 10^6^ conidia ml^−1^). The tubes containing seeds were sealed, para filmed and kept at 30 °C for 30 min in a shaking incubator. Control seeds were treated with 20 mL sterile water instead of a spore suspension and incubated under the same conditions. Inoculated seeds were left to dry in Petri dishes overlaid with Whatman No.1 filter paper overnight in a hood. Subsequently, the seeds were inoculated onto tissue culture media (MS) and germinated at 28 °C in a plant growth chamber, Conviron (Winnipeg, Manitoba, Canada). Growth was monitored for 14 days, then plant tissue (shoots and roots) were harvested and stored at −80 °C after being flash frozen in liquid nitrogen for DNA/RNA extraction.

### 5.5. DNA Extraction from Aspergillus flavus and Maize Tissues

DNA was extracted from 100 mg of each of the following samples: *A. flavus* KSM014 mycelia, infected and control healthy maize tissues following the method of [49] with modifications. Briefly, 2% SDS, 100 mM Tris pH 8.0, modified TES buffer, 10 mM EDTA, and 2% (*w*/*v*) polyvinylpyrrolidone (PVP) was prepared. Four hundred and fifty microliters of TES buffer and 5 µL RNase (10 mg/mL) was added to a 2 mL microtube containing the tissues and homogenised with a microtube pestle or vortex for 15 min. 20 µL Proteinase K (1 µg/µL) was added, vortexed for 1 min, and then incubated at 60 °C for 1 h. 160 µL of 5 M NaCl (0.3 vol.), 70 µL 10% CTAB (0.1 vol.) was then added and subsequently incubated for 10 min at 65 °C. Chloroform/isoamyl-alcohol (24:1) (750 µL) was added, vortexed for 5 min and again incubated for 30 min on ice and then centrifuged for 10 min at 14,000 rpm. The aqueous phase was transferred to a new 2 mL microtube, 300–350 µL isopropanol (0.55 vol.) added and then mixed gently for 30 s and left to stand at room temperature (RT) for 30 min. The mixture was centrifuged at 14,000× *g* rpm for 10 min. The supernatant was discarded, the pellets rinsed twice with chilled 70% ethanol (700 µL), gently mixed without and centrifuged again for 2 min at 14,000× *g* rpm. Ethanol was discarded, pellets air dried and then dissolved in 40 µL TE buffer (10 mM Tris-Cl pH 8, 1 mM EDTA pH 8.0) or nuclease free water. DNA integrity was assessed on a 1% agarose/EtBr gel and the concentration quantified on a Nano-Drop^TM^ 1000 spectrophotometer (NanoDrop Technologies, Silverside Rd, Wilmington, USA). DNA was diluted to 10 ng/µL for further analysis. The chemicals used were purchased from (Sigma-Aldrich, Spruce St., St Louis, USA).

### 5.6. Primer Design 

Three sets of primers (Table 2); *β-tubulin*, *Elongation factor 1 alpha* (*Ef1α*) and Membrane protein (MEP) were used in this study. *β-tubulin* was designed in Primer3 ver. 4.0 programme [50], whereas, *Ef1α* and MEP were obtained from Dr. Shane Murray, MCB lab. 227 (pers. Comm). Potential secondary structure formation was assessed in DNAMAN software ver. 6.03 (Lynnon LLC., San Ramon, CA, USA, 2015) and further verified in OligoAnalyzer Tool (Integrated DNA Technologies). The PCR and melt curve analysis were used to identify both specific and non-specific amplification. 

### 5.7. PCR Amplification

Conventional PCR amplification was performed in a volume of 25 µL and consisted of 10× reaction buffer, with MgCl_2_, 0.5 µL of 10 μM dNTPs (Bioline), 1 µL of 10 μM reverse and forward primers, 1 µL of 10 ng DNA template, 0.2 µL Kapa Taq and sterile water. Cycling conditions were performed according to the following protocol: 1 cycle at 94 °C for 5 min followed by 35× (at 94 °C for 30 s, at 60 °C for 45 s, at 72 °C for 90 s). A final elongation step was for 7 min at 72 °C and the reaction mixture was held at 4 °C until analysed. The PCR products were assessed on a 2% agarose/EtBr gel in 1 X TAE buffer (Tris–acetate 40 mM and EDTA 1.0 mM). Fermentas 100 bp DNA ladder was used as a molecular size marker.

### 5.8. Aflatoxin Standards, Standard Curves and Fungal Quantification 

The mycotoxin reference standards; aflatoxin B + G mixture dry concentrate was prepared and stored in a freezer at −20 °C according to manufactures recommendations (Sigma Aldrich, Germany). A working stock solution for bioanalytical analysis was prepared in a one-fold dilution containing 200 µg L^−1^ AFB1, 50 µg L^−1^ AFB2, 200 µg L^−1^ AFG1 and 50 µg L^−1^ AFG2 and intermediate solutions stored in amber bottles at −20 °C for three months and/or −80 °C for longer storage.

A ten-fold serial dilution of pooled 10 ng genomic DNA extracts from control plants and *A. flavus* were used to generate standard curves. For each dilution, the threshold cycle (Ct) values were plotted against the logarithm of the starting quantity of the template. Efficiencies of amplification were generated from the slopes of the standard curves slopes [53,54]. Linear regression curves were drawn, and the qPCR efficiency calculated as:(1)$$E=10\left(\frac{-1}{Slope}\right).$$

The amount of target DNA in an unknown sample was extrapolated from the respective standard curves.

Isolated DNA (10 ng) from healthy and infected maize shoots and roots were used to test the specificity of the primers. To exclude false negative results, the template DNA samples from fungi were tested for PCR amplification with primer pairs *β-tub* and *EF1*α. DNA from control plant tissues and pure fungal cultures (*A. flavus*) were pooled together, diluted to 10 ng/µL and used to estimate the amount of fungal DNA template in the infected plant tissue. The final fungal DNA template concentrations were 1, 5 × 10^−1^, 2.5 × 10^−1^, 1.25 × 10^−1^, 6.25 × 10^−2^, 3.125 × 10^−2^ ng/µL. These dilutions were used to determine the detection limits of the *β-tub* and *EF1*α primer pair in the infected plant tissues. A serial dilution of DNA extracted from healthy maize tissue also was prepared to measure the detection limits of the MEP. To normalise the gene quantification between different samples, the amount of fungal DNA as calculated by the Ct value for *β-tub* and/or *EF1α* was divided by the amount of maize DNA as calculated by the Ct values for MEP. Rotor Gene 6000 2 plex HRM (Corbett Life Science Research, Mortlake, Australia) was used to evaluate the gene expression profiles. Master mix, Kapa SYBR Fast Kit (Kapa BioSystems, Cape Town, South Africa), containing DNA polymerase, dNTPs, reaction buffers and 3 mM MgCl_2_ were used for each PCR reaction. Final concentrations of 1× Kapa SYBR green, 10 μM gene specific primers (0.2 µL forward and 0.4 µL reverse), and 1 µL of DNA template were prepared to a total volume of 20 µL, using nuclease-free water. Primer sets of specific genes (Table 2) were used in separate reactions which were performed in triplicate.

For assessing the integrity and quality of the isolated DNA, samples from control and infected tissues of the plant, and saprophytic fungi were subjected to PCR analysis with the reference genes under the following amplification conditions: 95 °C for 10 min; 35 cycles of 95 °C for 3 s, 64 °C for 20 s, 72 °C for 1 s for MEP and at Ta 62 °C for both *β-tub* and *Ef1α.*

### 5.9. Metabolite Extraction

Fungal metabolites were extracted from *A. flavus* strains using different solvents. Briefly, fresh mycelia (200–400 mg) was scraped off the culture plate and placed into a screw capped disposable vials containing, and/or not approximately four glass beads of 4 mm in diameter (Merck KGaA, Darmstadt, Germany). Aflatoxins and other metabolites were extracted ultrasonically for 15 min by using, 2–10 mL of extraction solvent consisting of methanol dichloromethane, ethyl acetate (MeOH:DCM:EtOAc (1:2:3)) in 1% formic acid. Extracts were centrifuged at 14,000× *g* rpm for 15 min at 4 °C and 500 µL transferred to sterile 2 mL tubes. The samples were dried with a Savant SpeedVac Plus SC210A Concentrator (Thermo Scientific, Ramsey, MN, USA) for 12 h. The residue was reconstituted in 400 µL methanol with 0.6% (*v*/*v*) FA, 0.02% (*v*/*v*) HCl and 2.5% (*v*/*v*) water. The reconstitution was carried out in an ultrasonic bath sonicator for 10 min at RT. Samples were centrifuged at 14,000× *g* rpm for 15 min at 4 °C and 250 µL transferred to glass vials for TLC and HPLC analysis. 

### 5.10. Thin Layer Chromatography

Thin Layer Chromatography (TLC) was carried out on a TLC silica gel 60 plate 20 × 20 cm (Merck, KGaA, Darmstadt, Germany). Acetonitrile/methanol/formic acid (9:1:0.2 *v*/*v*) was used as the mobile phase. Ten microliters of the aflatoxin standard mix containing 5.86 µg/mL of AFB1 and AFG1, and 1.70 µg/mL of AFB2 and AFG2 and 20 µL of test samples were spotted on TLC plates and run for 70–90 min in a TLC tank at RT. The plates were left to air dry in the fume hood at RT for approximately 30 min. Dried plates were either observed under UV light (wavelength 254 and 366 nm), or sprayed with *p*-anisaldehyde solution, vanillin solution or exposed to an iodine vapour. The intensity of the sample spots against the standard aflatoxins were compared and aflatoxins concentrations calculated based on the equation:(2)$$E=\frac{SCV}{WZ}$$
where; *E* = aflatoxins (µg/kg)*S* = µL of aflatoxin standard equal in fluorescence to sample spot*C* = aflatoxin standard concentration in µg/mL*V* = final dilution of the sample extract (µL)*Z* = sample matching the standards (µL)*W* = sample extract weight (mg)

### 5.11. High Performance Liquid Chromatography and Optimisation 

Optimisation and chromatographic separations were achieved on an Agilent HPLC 1200 system, comprised of a binary pump equipped with micro vacuum degasser, thermostatic auto sampler, column compartment, and Diode Array Detector (Agilent Technologies, Waldbronn, Germany). Fluorescence detection was performed at excitation and emission wavelengths from 200 to 410 nm. Separations were performed on Agilent Zorbax Eclipse XDB-C_18_ column, 4.6 × 150 mm I.D., particle size 5 µm (Agilent Technologies, Waldbronn, Germany), maintained at 40 °C operating at a flow rate of 1.0 mL min^−1^. Water and acetonitrile, both containing 0.005% trifluoro acetic acid (TFA) were used as mobile phases. A gradient starting from 85% water and 15% acetonitrile used to 100% acetonitrile for 20 min maintained at 100% acetonitrile for 23 min and final 15% acetonitrile for 27 min. Sample injection volume was 15 µL. All chemicals used were HPLC grade. UV wavelength signals were set at 200, 210, 230, 270, 280, 320, 350, and 410 nm. Aflatoxins in the sample solution were identified by comparison of their retention times and peak height/area with corresponding standards in the standard solution.

### 5.12. In Vitro Co-Infection of Maize Lines and Biocontrol Strategy

Undamaged kernels from KDV1 and GAF4 maize lines were surface sterilised in hot water for 45 s at 80 °C, as previously described [55]. Briefly, maize moisture content was quantified with a HB43 Halogen Moisture Analyzer (Mettler Toledo, Columbus, OH, USA), and adjusted to 25% by soaking the kernels in sterile water for 30–60 min, as described by [56]. Approximately 30 sterilized grains were seeded with aliquots of spore suspensions of 1000 µL (1 × 10^6^ conidia/mL) in a level 2 biosafety cabinet with respective fungal isolates (atoxigenic and aflatoxigenic) at different ratios (0:100, 25:75, 50:50, 75:25, 100:0) in sterile vials. The controls were inoculated with 1000 µL sterile water instead of fungal spore suspension. Both infected and control vial contents were shaken for 30 s in a vortex mixer (SciQuip Ltd., Shropshire SY4 5NU, UK) to ensure complete and uniform coating of kernels with inoculum. Vial lids were loosened briefly, to enable gas exchange and incubated at 30 °C for 14 days in the dark. At the end of the incubation period, fungal activity was discontinued by addition of 50 mL, 80% methanol or halted by oven drying at 45 °C for one day and the contents then prepared for aflatoxin extraction and further analysis. Three biological replicas of the experiments were performed twice. The efficiency of surface sterilization, and the ability of kernels to germinate, were monitored by plating five randomly selected kernels from each vial onto a selective YES and YEP media, followed by incubation at 30 °C in the dark for 14 days. Approximately, 99% of the kernels germinated and no fungal contaminants were observed at the end of the incubation period.

### 5.13. Statistical Analysis

The statistical analysis was performed as previously described [33]. Aflatoxin concentration and percentage reduction were log transformed prior to analyses using GraphPad Prism, One-way analysis of variance (ANOVA), Tukey’s Multiple Comparison Test (TMCT) and post-test for linear trend analysis. Mean differences in aflatoxin levels (percent difference between inoculated maize and control maize treatments) were calculated as:(3)$$1-\frac{TACM\;co-inoculted\;with\;both\;atox\;and\;aflatox\;isolates\;of\;A.\;\;flavus\;}{TACM\;inoculated\;with\;the\;aflatoxigenic\;isolate\;alone}\times 100$$
where: TACM is total aflatoxin content in maize; atox: atoxigenic and aflatox: aflatoxigenic.

Standard deviations of mean differences in aflatoxin levels were calculated as a measure of variability in efficacy. The efficiency (E) of each isolate was calculated as:(4)$$E=\frac{R}{\frac{A}{A+T}}$$
where R is the percentage of aflatoxin reduction and the denominator is the percentage of the total *A. flavus* inoculum made up by the atoxigenic isolate (A). ‘A’ is the quantity of atoxigenic strain and ‘T’ is the quantity of aflatoxin-producer. All analyses and calculations were performed in GraphPad Prism software ver. 5.0.2.

## Figures and Tables

**Figure 1 toxins-11-00179-f001:**
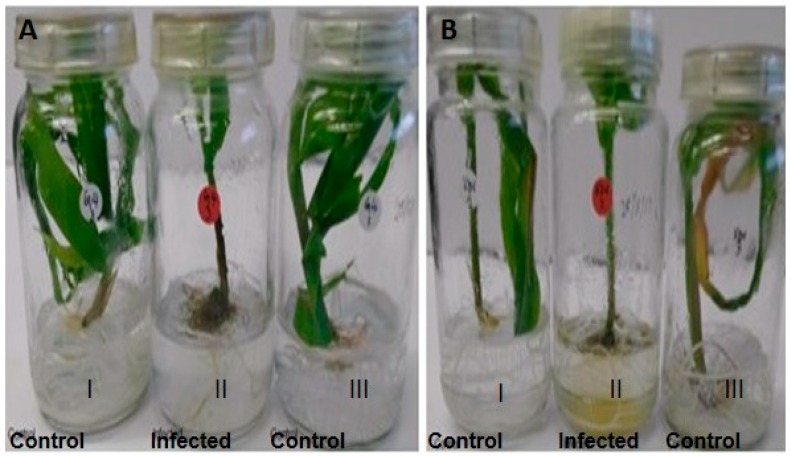
The respective GAF4 (**A**) and KDV1 (**B**) maize lines with and without *Aspergillus flavus* KSM014 infection. The uninfected maize control plants (I & III) and the infected plants (II) are shown 14 days after germination.

**Figure 2 toxins-11-00179-f002:**
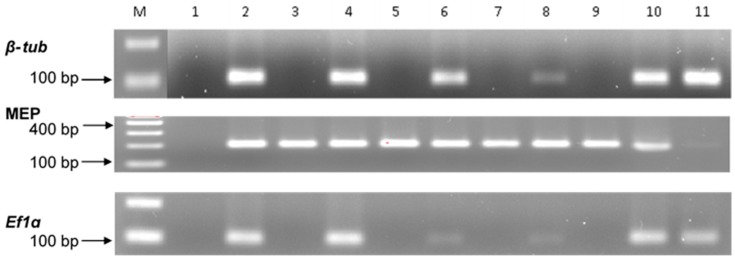
Gel electrophoresis of qPCR amplicons for *Aspergillus flavus* maker genes (*β-tub*, *Ef1α*) and maize marker gene (MEP) assessed on 2% agarose/EtBr gel run at 80 volts for 45 min. **M**. 100 bp ladder; **1**. NTC; **2**. Pooled samples (Pure fungal gDNA and maize gDNA); **3**. GAF4 (control roots); **4**. GAF4 (infected roots) **5**. GAF4 (control shoots); **6**. GAF4 (infected shoots); **7**. KDV1 (control shoots); **8**. KDV1 (infected shoots); **9**. KDV1 (control roots); **10**. KDV1 (infected roots); **11**. KSM014 (Positive *A. flavus* control).

**Figure 3 toxins-11-00179-f003:**
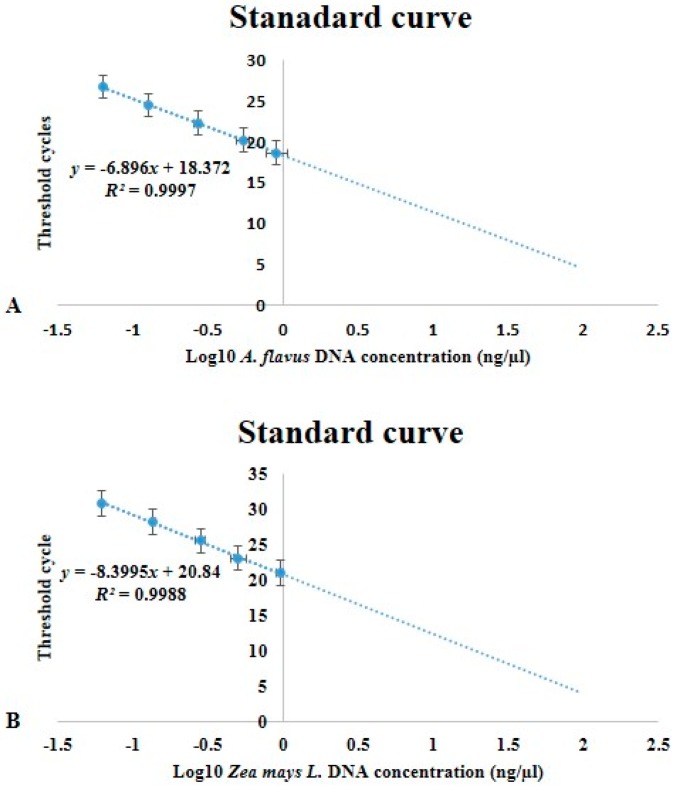
Standard curves used for Real time quantitative polymerase chain reaction (RT-qPCR) SYBR green assay to quantify *A. flavus* in maize plant tissues. The curves illustrate the linear regression, efficiency and sensitivity of qPCR for early fungal detection employing marker genes, *β*-*Tub* and *MEP* to a total of 10 ng genomic DNA. (**A**) standard curve for *β*-*Tub* amplification using *A. flavus* DNA diluted in maize carrier DNA (**B**). standard curve for MEP amplification from using serial dilutions of maize genomic DNA. (Error bars shows the standard deviations of the mean triplicate gDNA concentrations).

**Figure 4 toxins-11-00179-f004:**
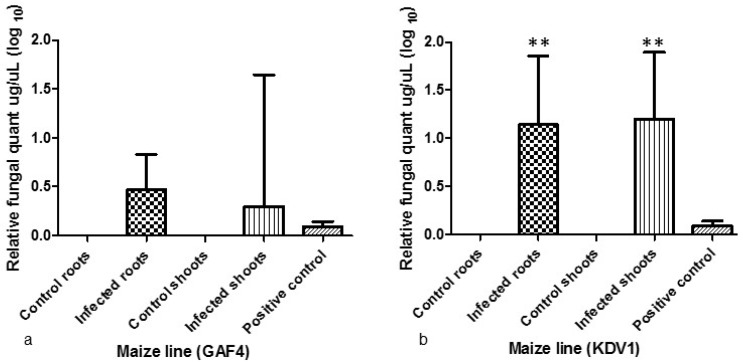
qPCR analysis showing fungal load of *A. flavus* KSM014 in the root and shoot tissue of GAF4 and KDV1 maize lines respectively. Fungal biomass was measured in infected and non-infected (control) GAF4 (**a**) and KDV1 (**b**) maize lines within 14 days where the *A. flavus β-tub* gene was used for fungal quantification against the maize MEP gene. A one-way ANOVA and Tukey’s Multiple Comparison Test revealed (*p* < 0.05). Asterisks indicate significance and the error bars shows standard mean deviation (*n* = 3).

**Figure 5 toxins-11-00179-f005:**
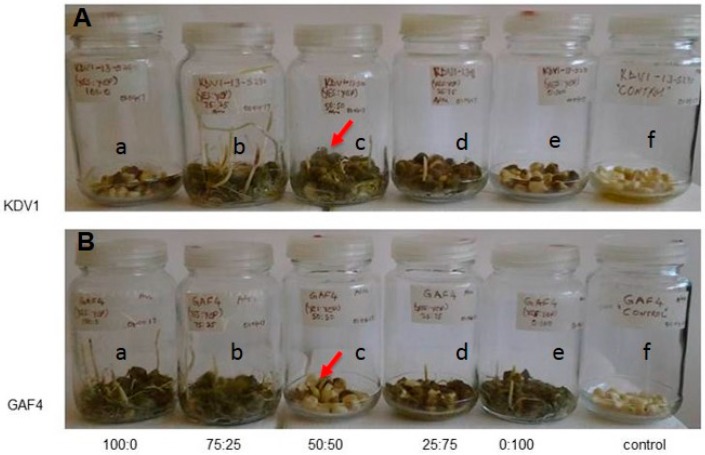
Biocontrol approach with atoxigenic KSM012 and aflatoxigenic KSM014 strains of *Aspergillus flavus* to mitigate aflatoxin production. The kernels for sensitive (**A**; KDV1) and resistant (**B**; GAF4) maize lines were co-infected at different ratios (0:100; 25:75; 50:50; 75:25; 100:0) with atoxigenic and aflatoxigenic strains.

**Figure 6 toxins-11-00179-f006:**
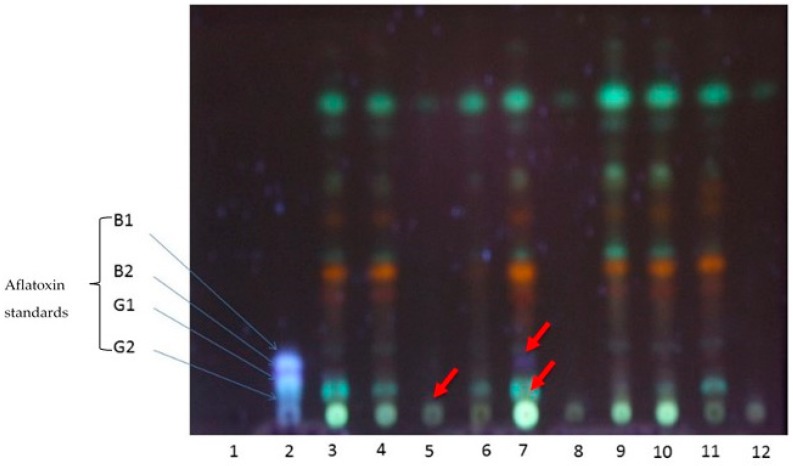
Thin layer chromatography plates showing the presence or absence of mycotoxins from the isolates in comparison with the aflatoxin standards at long wavelength, 365 nm. Lanes: (1) Blank; (2) Standard; atoxigenic KSM012 and aflatoxigenic KSM014 ratios for the GAF4 line (G) were (3) G100/0; (4) G75/25; (5) G50/50; (6) G25/75; (7) G0/100; and (8) K100/0; (9) K75/25; (10) K50/50; (11) K25/75; (12) K0/100 for the KDV1 line (K). (B1: aflatoxin AFB1; B2: aflatoxin AFB2; G1: aflatoxin AFG1; G2: aflatoxin AFG2)**.**

**Figure 7 toxins-11-00179-f007:**
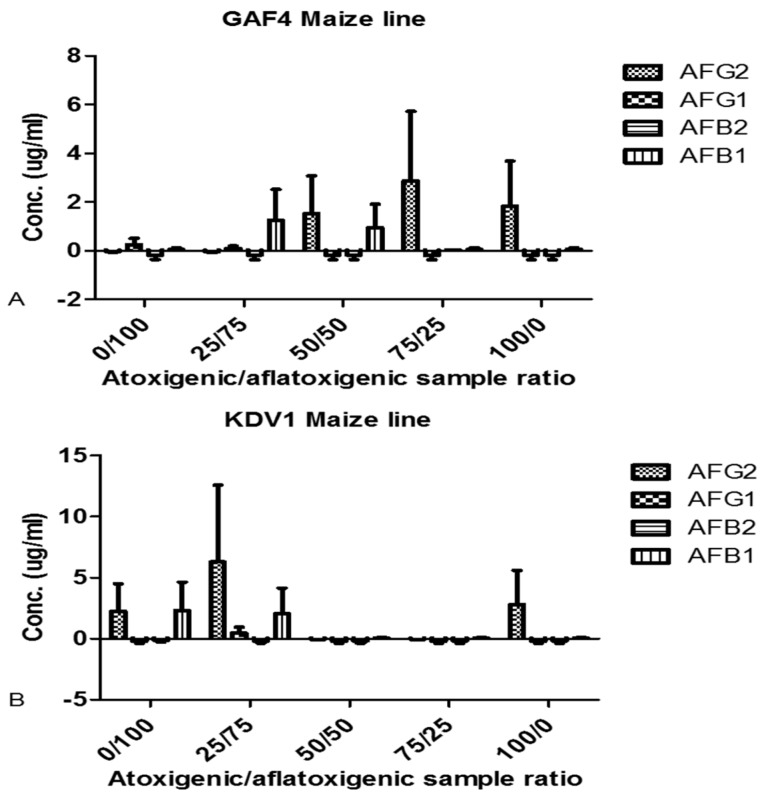
High Performance Liquid Chromatography analysis highlighting biocontrol of aflatoxigenic (KSM014) by atoxigenic (KSM012) *Aspergillus flavus* when co-infected at different ratios in maize lines GAF4 (**A**) and KDV1 (**B**) respectively. (AFB1: aflatoxin B1; AFB2: aflatoxin B2; AFG1: aflatoxin G1; AFG2: aflatoxin G2)**.** A Tukey’s multiple comparison test revealed that AFG2 versus AFG1 and AFG2 versus AFB2 for the GAF 4 maize line were significant (*p* < 0.05).

**Table 1 toxins-11-00179-t001:** Phenotypic characteristic measurements of control and infected (GAF and KDV1) maize lines with *Aspergillus flavus* isolate KSM014 (*n* = 3) taken after 14 days of growth. Massive variation was observed in roots and shoots of both maize lines, with KDV1 exhibiting more severe symptoms of stunted growth.

Phenotypic Characteristic																
Maize Line	Control Roots				Control Shoots				Infected Roots				Infected Shoots			
	Exp 1	Exp 2	Exp 3	Average (mm)	Exp 1	Exp 2	Exp 3	Average (mm)	Exp 1	Exp 2	Exp 3	Average (mm)	Exp 1	Exp 2	Exp 3	Average (mm)
GAF4	285	260	278	274.33	352	322	312	328.67	134	113	98	115	142	185	111	146
KDV1	272	252	232	252	344	300	323	322.33	78	83	84	81.67	82	87	91	86.67

**Table 2 toxins-11-00179-t002:** Specific primers used for total fungal and strain quantification.

Primer Name	Forward Primer (5′-3′)	Reverse Primer (5′-3′)	Product Size (bp)	*Ta*	Reference
Membrane Protein (MEP)	TGTACTCGGCAATGCTCTTG	TTTGATGCTCCAGGCTTACC	203	64 °C	Manoli et al. [51]
Elongation Factor 1 alpha (*EF1α*)	CGTTTCTGCCCTCTCCCA	TGCTTGACACGTGACGATGA	102	62 °C	Nicolaisen et al. [52]
*β-Tubulin*M	TCTTCATGGTTGGCTTCGCT	CTTGGGTCGAACATCTGCT	118	62 °C	Mitema et al. [33]

******Ta:* annealing temperature ***** bp: base pair

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
