# Peer review of "The Development of a qPCR Assay to Measure Aspergillus flavus Biomass in Maize and the Use of a Biocontrol Strategy to Limit Aflatoxin Production"

_toxins, 2019, doi:10.3390/toxins11030179_

Round 1
Reviewer 1 Report
The topic of the study, Aspergillus flavus-maize pathosystem fits well in the scope of the journal as the toxin produced by the fungus is severely threatening the food security escpecially in developing countries. The manuscript presents methodological development of a molecular assay, which is finally used in studying the effect of biocontrol against alfatoxicoses. To my knowledge, the methodology is standard and the experiments with adequate controls are appropriate for the purpose. Also the language is fluent scientific English and the text reads well. The list of references seems to cover relevant literature including recent papers. Alltogether, technically the MS is very solid.
The introduction of the manuscript gives information, which is tightly linked to the research aims. I would like to see also a short, more general description of the problem; fungus-plant pathosystem as well as importance of the alfatoxicose problem for maize production escpecially in developing countries. Information of the geographical distribution of alfatoxicoses in Kenya appears to be an important background information, but it is introduced too late, in the discussion.
After reading the introduction, it is not clear why to focus on the study aims. It is noteworthy that the study aims are introduced: “study was divided in four parts”. Accordingly, the study remains descriptive in the absence of specific study aims and hypotheses.
The conclusion of observation of susceptibility in KDV1 maize is a major finding as it appears to explain the frequent alfatoxicosis in Makueni area. However, this important background information is not in the introduction. The conclusions are mostly well based on the results.
The major weakness of the MS is that the reason or need to develop a new qPCR assay is not really explained. Also methodological benefits of the new assay over the previously described assays are not clear and should be further clarified. Now the MS is built on qPCR results, but as the introduction states, similar studies have already been made. Perhaps the assay and analyses lacking apparent novelty should not be in focus?
My suggestion is that the manuscript should be rewritten with more problem-based approach. The beginning of the introduction should describe problem generally, then rising the issue of frequent alfatoxicoses in Kenya, and finally present the study aims as potential solutions for the problem. I suppose the rest of the manuscript could be easily modified to correspond the introduction.
Author Response
Reviewer # 1
The topic of the study, Aspergillus flavus-maize pathosystem fits well in the scope of the journal as the toxin produced by the fungus is severely threatening the food security especially in developing countries. The manuscript presents methodological development of a molecular assay, which is finally used in studying the effect of biocontrol against alfatoxicoses. To my knowledge, the methodology is standard and the experiments with adequate controls are appropriate for the purpose. Also the language is fluent scientific English and the text reads well. The list of references seems to cover relevant literature including recent papers. Altogether, technically the MS is very solid.
Point 1: The introduction of the manuscript gives information, which is tightly linked to the research aims. I would like to see also a short, more general description of the problem; fungus-plant pathosystem as well as importance of the alfatoxicose problem for maize production especially in developing countries. Information of the geographical distribution of alfatoxicoses in Kenya appears to be an important background information, but it is introduced too late, in the discussion.
Response 1: We have included the short description, fungus-plant pathosystem and the importance of aflatoxicoses in developing countries. We have reorganized and introduced aflatoxicoses in Kenya in the first paragraph. The discussion and conclusion has also been reorganised as per the reviewer’s suggestions.
Point 2: After reading the introduction, it is not clear why to focus on the study aims. It is noteworthy that the study aims are introduced: “study was divided in four parts”. Accordingly, the study remains descriptive in the absence of specific study aims and hypotheses.
Response 2: We have re-written that section, as noted below:
“This study investigated whether the respective maize lines KDV1 and GAF4, grown in different regions of Kenya, contribute to increasing or limiting the biomass of KSM014, an aflatoxigenic strain. In this regard we developed and tested a qPCR assay to quantify the amount of A. flavus biomass in infected maize tissues. Furthermore we tested if the atoxigenic A. flavus strain KSM012, (NCBI¬_accession MG385137) could serve as a biocontrol agent to minimise aflatoxin contamination of maize kernels colonised by the toxigenic A. flavus strain KSM014, (NCBI_accession MG385138).”
Point 3: The conclusion of observation of susceptibility in KDV1 maize is a major finding as it appears to explain the frequent alfatoxicosis in Makueni area. However, this important background information is not in the introduction. The conclusions are mostly well based on the results.
Response 3: We have included the background information in the introduction of the manuscript as suggested.
Point 4: The major weakness of the MS is that the reason or need to develop a new qPCR assay is not really explained. Also methodological benefits of the new assay over the previously described assays are not clear and should be further clarified. Now the MS is built on qPCR results, but as the introduction states, similar studies have already been made. Perhaps the assay and analyses lacking apparent novelty should not be in focus?
Response 4: Indeed, qPCR assays have been used previously but specifically we used different marker genes (MEP for maize, β-tub and Ef1ɑ for A. flavus) and SYBR green dye compared to other studies (Coninck et al., 2012; Sanzani et al., 2014; Mylroie et al., 2016). As far as we are aware, this is the first report of using a RT-qPCR assay to detect A. flavus in Kenya. We have highlighted this on lines 99-101 (section 2.1, last paragraph) and in discussion section, lines 215-220.
Point 5: My suggestion is that the manuscript should be rewritten with more problem-based approach. The beginning of the introduction should describe problem generally, then rising the issue of frequent alfatoxicoses in Kenya, and finally present the study aims as potential solutions for the problem. I suppose the rest of the manuscript could be easily modified to correspond the introduction.
Response 5: We have reorganised the manuscript as suggested by the reviewer.
Reviewer 2 Report
I think that the paper is quite confusing. There are lots of information mixed and not well organized that do not allow to really understand methodology and results obtained.
Specific remarks are:
References are not linked correctly in the text. I found lots of discrepancies and the number is not correct. In the paper are reported references up to number 49 and in the reference list there are only 48 references. It is quite hard to read a text that refer to wrong papers.
Experimental design: I think that it is confusing. In the results section seem quite clear that 1 strain of A. flavus able to produce aflatoxin was used (KSM014) as well as 1 atoxigenic strain but in the materials and methods section you speak about 5 different fungal strains (KSM012, KSM014, HB021, HB026 and HB027) without explaining if they are aflatoxin producers or not aflatoxin producers. Moreover I do not understand the methodology used to obtain spores (using 2 different media and mixing spores of different fungi).
You did not mention the number of samples you prepared in all the different types of experiments you did.
Did you work with replicates?
In the experiment for biocontrol strategies you speak about higher and lower levels of colonisation and different rates of fungal colonisation but how did you measured it? Did you use any scale? Are differences statistically significative?
Regarding DNA extraction: all material used has to report supplier (brand).
Regarding DNA extraction, line 330: why did you used room temperature? Usually low temperature is required.
Use of atoxigenic strain to control aflatoxin production: I think that the choice of 1 atoxigenic strain needs many work. First of all it is necessary to transfer it successively many times (20-30 times) in order to define if it is really not able to produce aflatoxin and than VCG group determination is absolutely necessary. I think that to use only 1 atoxigenic strain and compare with only 1 toxigenic strain is not sufficient to determine that it can be a good strain for biocontrol.
Author Response
Reviewer ## 2
Point 1: There are lots of information mixed and not well organized that do not allow to really understand methodology and results obtained.
Response 1: We have noted and reorganised the sections as suggested by reviewer 1.
Point 2: References are not linked correctly in the text. I found lots of discrepancies and the number is not correct. In the paper are reported references up to number 49 and in the reference list there are only 48 references. It is quite hard to read a text that refer to wrong papers.
Response 2: Thank you for noting this, we have followed up and made the required updates and addition of references where necessary.
Point 3: Experimental design: I think that it is confusing. In the results section seem quite clear that 1 strain of A. flavus able to produce aflatoxin was used (KSM014) as well as 1 atoxigenic strain but in the materials and methods section you speak about 5 different fungal strains (KSM012, KSM014, HB021, HB026 and HB027) without explaining if they are aflatoxin producers or not aflatoxin producers. Moreover I do not understand the methodology used to obtain spores (using 2 different media and mixing spores of different fungi).
Response 3: This specific section is where different strains and media were used for the metabolite analysis. There were two types of media to culture the isolates; YES-aflatoxin inducing media and YEP-non-inducing media. The A. flavus strain KSM014 was toxigenic while strains KSM012, HB021, HB026 and HB027-were atoxigenic. This has been explained in section 5.1, lines 282-289.
Point 4: You did not mention the number of samples you prepared in all the different types of experiments you did. Did you work with replicates?
Response 4: Kindly note that this has been mentioned in lines 317; 321-322. We worked with triplicates-refer to lines 317; 321-322 and three biological replicates of the experiments were performed twice, refer to lines 460.
Point 5: In the experiment for biocontrol strategies you speak about higher and lower levels of colonisation and different rates of fungal colonisation but how did you measured it? Did you use any scale? Are differences statistically significative?
Response 5: This was done first visually by looking at the amount of colonisation based on triplicate experiments. Visual assessment scoring scale was established (0% for no fungal colonisation, 50% for half kernel fungal colonisation and 100% for entire fungal kernel colonisation). This was then validated using TLC and HPLC analysis with HPLC analysis providing statistically significant results, refer lines 189-191.
Point 6: Regarding DNA extraction: all material used has to report supplier (brand).
Response 6: We have noted and inserted the relevant suppliers/brands.
Point 7: Regarding DNA extraction, line 330: why did you used room temperature? Usually low temperature is required.
Response 7: Kindly refer to lines 329-330; the storage was at -80 ° C after being flash frozen in liquid nitrogen
Point 8: Use of atoxigenic strain to control aflatoxin production: I think that the choice of 1 atoxigenic strain needs many work. First of all it is necessary to transfer it successively many times (20-30 times) in order to define if it is really not able to produce aflatoxin and than VCG group determination is absolutely necessary. I think that to use only 1 atoxigenic strain and compare with only 1 toxigenic strain is not sufficient to determine that it can be a good strain for biocontrol.
Response 8: We agree with the reviewer and noted the recommendation in the discussion and conclusion which we have re-written accordingly (see lines 277-288 and 301-303)
Reviewer 3 Report
The manuscript “The development of a qPCR assay to measure Aspergillus flavus biomass in maize and use of a biocontrol strategy to limit aflatoxin production” is well designed study which deals with biocontrol strategy to limit aflatoxin production. In my opinion, the claims of the study are fully supported by the experimental data. However, the weak part of the study is technical presentation of results. It seems like figures are not of the good quality and it should be fixed. There is a lot of empty space in the manuscript. Also, Discussion seems to be poorly written. English language and style used should be improved. Despite to that, the topic of the paper is very important.
Author Response
Reviewer ### 3
The manuscript “The development of a qPCR assay to measure Aspergillus flavus biomass in maize and use of a biocontrol strategy to limit aflatoxin production” is well designed study which deals with biocontrol strategy to limit aflatoxin production.
Point 1: In my opinion, the claims of the study are fully supported by the experimental data. However, the weak part of the study is technical presentation of results. It seems like figures are not of the good quality and it should be fixed. There is a lot of empty space in the manuscript. Also, Discussion seems to be poorly written. English language and style used should be improved. Despite to that, the topic of the paper is very important.
Response 1: Based on the comments of the other reviewers we have made changes to the presentation of the technical results. We have replaced the figures (Fig.1) and the spaces have been deleted and reorganised. We have reorganised and rewritten the sections as suggested by reviewers. Lastly, we have used the American English version style and run the spelling check on the entire manuscript and corrected where necessary.
Round 2
Reviewer 1 Report
The authors have reorganised the manuscript and added requested information. Now the MS appears as problem based case study and my suggestion is that MS could be published.
Author Response
Point 1: The authors have reorganised the manuscript and added requested information. Now the MS appears as problem based case study and my suggestion is that MS could be published.
Response 1: Thank you for the review and suggestion that MS can be published
Reviewer 2 Report
The paper has been improved. I still remain confused on cultures of fungi description. I still not understand why you cultivated 5 fungal strains and than comment only 1 toxigenic and 1 atoxigenic. Why did you cultivated more strains? I think that you have to refer only to the 2 strains you effectively used in the experiments (KSM014 e KSM012) and delete referements to the other 3 (HB0212, HB026, HB027). Did you use a mixed inoculum with the 5 strains to inoculate the plants? If yes, why? Please justify it.
Author Response
Point 1: The paper has been improved. I still remain confused on cultures of fungi description. I still not understand why you cultivated 5 fungal strains and than comment only 1 toxigenic and 1 atoxigenic. Why did you cultivated more strains? I think that you have to refer only to the 2 strains you effectively used in the experiments (KSM014 e KSM012) and delete referements to the other 3 (HB0212, HB026, HB027). Did you use a mixed inoculum with the 5 strains to inoculate the plants? If yes, why? Please justify it.
Response 1: We have deleted the three fungal strains HB021, HB026 and HB027 as suggested.